# Mechanical and Tribological Performance of HDPE Matrix Reinforced by Hybrid Gr/TiO_2_ NPs for Hip Joint Replacement

**DOI:** 10.3390/jfb14030140

**Published:** 2023-03-02

**Authors:** Ahmed Nabhan, Galal Sherif, Ragab Abouzeid, Mohamed Taha

**Affiliations:** 1Production Engineering and Mechanical Design, Faculty of Engineering, Minia University, El-Minia 61111, Egypt; 2Cellulose and Paper Department, National Research Centre, 33 El-Buhouth Street, Dokki, Giza 12622, Egypt; 3School of Renewable Natural Resources, Louisiana State University AgCenter, Baton Rouge, LA 70803, USA; 4Mechanical Engineering Department, College of Engineering and Technology, Arab Academy of Science, Technology and Maritime Transport, Sadat Road, Aswan P.O. Box 11, Egypt

**Keywords:** HDPE nanocomposites, TiO_2_ nanoparticles, graphene, wear resistance, hip joint replacement

## Abstract

Hip joint collapse is a very common health problem. Many cases need a joint replacement, so nano-polymeric composites are an ideal alternative solution. Due to its mechanical properties and wear resistance, HDPE might be considered a suitable alternative to frictional materials. The current research focuses on using hybrid nanofiller TiO_2_ NPs and nano-graphene with various loading compositions to evaluate the best loading amount. The compressive strength, modules of elasticity, and hardness were examined via experiments. The COF and wear resistance were evaluated via a pin-on-disk tribometer. The worn surfaces were analyzed based on 3D topography and SEM images. The HDPE samples with various compositions of 0.5%, 1.0%, 1.5%, and 2.0 wt.% filling content of TiO_2_ NPs and Gr (with a ratio of 1:1) were analyzed. Results revealed that hybrid nanofiller with a composition of 1.5 wt.% exhibits superior mechanical properties compared to other filling compositions. Moreover, the COF and wear rate decreased by 27.5% and 36.3%, respectively.

## 1. Introduction

Polymers are available, economical, corrosion-resistant, easy to manufacture, and biocompatible. These distinctive features have made polymers the focus of much research to enhance and develop their properties [1,2,3]. Polymers and their nano-based composites have been widely adopted in various biomedical fields. Total artificial joint replacement is one of the major challenges that many people may suffer from. Joint wear and tear can significantly restrict movement [4,5,6]. Many nano-additives, such as Au, Ag, Al_2_O_3_, TiO_2_, CuO, graphene, carbon nanotubes, carbon nanofibers, and cellulose, were evaluated to enhance their mechanical, physical, and tribological properties. The uniform dispersion of nanofillers in the polymeric matrix contributes to significantly improving their characteristics [7,8,9,10,11]. Composite systems were performed, which led to an impact on their unique attributes compared to pure polymers.

Total joint replacement, or arthroplasty, was performed to replace a worn-out joint with an artificial joint, which is usually made of metal, ceramic, or polymer. Polymeric nanocomposites were developed and evaluated for use as a frictional material in artificial joints. This could be because polymeric nanocomposites have high biocompatibility, phase stability, and distinct wear resistance [12,13,14]. Polyethylene matrix has been extensively applied due to its distinct mechanical, physical, and tribological characteristics. Various types of fillers (fibers, platelets, or particles) were embedded in a polyethylene matrix to modify their features. Ultra-high molecular weight polyethylene, or UHMWPE, was preferred to adopt as a host matrix due to its excellent response to any filler content. Multi-walled carbon nanotubes (MWCNTs) were recommended as fillers in UHMWPE matrices [15,16]. UHMWPE/MWCNT nanocomposites with loading amounts of 0.5 and 1 wt.% were evaluated for wear resistance. The results indicated that the wear rate of nanocomposites was reduced compared with the pure UHMWPE matrix [17]. Furthermore, the UHMWPE matrix was reinforced with volume fractions of 1% by weight. MWCNT exhibits a good improvement in its mechanical properties. The young’s modulus was increased by 80% more than the neat sample [18]. Argon plasma treatment was applied to enhance the tribological properties of UHMWPE/MWCNT nanocomposites with filler contents of 0.5, 1.0, 1.5, and 2.0 wt.%. It can be found that the plasma-treated samples exhibit a significant improvement in the composite properties. It can be evident that increasing the treatment time contributes to enhancing the hardness, surface roughness, and friction coefficient [19]. Effective loading content of 3 wt.% Al_2_O_3_ dispersed into a UHMWPE matrix contributes to improved wear resistance and mechanical properties [20].

The hybrid filling system has been adopted to take full advantage of the attributes of both fillers and increase effectiveness. Based on this, many studies have resorted to verifying this system to develop the properties of the matrix [21,22,23]. Hybrid nanofillers of hydroxyapatite—zirconia were recommended to evaluate the biological, mechanical, and wear performances of UHMWPE matrix. The results indicated that the zirconia incorporation contributes to enhanced biological behavior. Moreover, the yield strength of UHMWPE matrix reinforced with 2 wt.% of hydroxyapatite—zirconia, with an equal ratio, increases about 45% higher than the free sample, while the friction coefficient reduces up to 64%. This can be evident from the fact that the dispersion of hybrid nanofillers exhibits a favorable scale of enhancement [24]. Titanium oxide (TiO_2_) incorporated with hydroxyapatite (HA) was applied as hybrid nanofillers, which disperse in UHMWPE via a solvent mixture technique [25]. It was observed that loading contents of 3% TiO_2_ and 2% HA demonstrated their ability to carry out a superior level of improvement. Incorporation of carbon nanofibers (CNF) and paraffin oil with UHMWPE was demonstrated to assess the thermal, mechanical, and tribological behavior [26,27,28,29]. The hybrid nanocomposites exhibited a significant ability of wear resistance under both conditions, dry and wet. This may be due to its thermal features, self-lubricating effect, and low shear stress. Graphene-UHMWPE nanocomposite was prepared to explore the impact of incorporation of the graphene nanofiller in the UHMWPE matrix [30,31,32,33]. The results demonstrated that the presence of nano-graphene leads to the development of the thermal, mechanical, and tribological performances of UHMWPE matrix. This may be a consequence of its nature as a carbonate, which contributes to modifying and developing the composite’s characteristics. Few studies have adopted high-density polyethylene (HDPE) as a host matrix. Graphene oxide was dispersed in an HDPE/UHMWPE matrix to explore its impact on wear mechanisms. The results demonstrated that the friction coefficient and wear rate were significantly reduced. It can be evident that the presence of graphene oxide is a key factor in developing the tribological performance of HDPE/UHMWPE composites [34,35]. The dispersion of 1 wt.% of the nanodiamond in an HDPE matrix was performed. The finite element model of the knee joint was established to evaluate the equivalent stresses and predict contact pressure with various flexion angles [36]. Hybrid nanofiller MWCNTs and boron nitride nanoplatelets (h-BNNPs), were used to amalgamate into an HDPE matrix. The results confirmed that loading contents of 0.25 and 0.15 wt.% of MWCNTs and h-BNNPs, respectively, were key factors in improving the mechanical and thermal performance of HDPE [37]. Therefore, an FE model of the hip joint was constructed to confirm HDPE nanocomposite as an alternative material for a made-up hip prosthesis [38]. The simulation results confirmed that the generated stresses for HDPE nanocomposites, with dispersion of 0.25% of MWCNTs and 0.15% of h-BNNPs, were within the permissible limit. Low loading content of Al_2_O_3_ nanoparticles exhibited a good reaction to developing the tribological and mechanical characteristics of HDPE matrix [39]. This is a consequence of the good dispersion of the Al_2_O_3_ NPs, which was attributed to improving the load carrying ability and matrix structure. Moreover, the HDPE/MWCNT nanocomposites were examined to identify the MWCNT impact on chain structure [40]. It can be demonstrated that wear performance and mechanical properties were enhanced, which may be due to the successful incorporation of MWCNT into the HDPE matrix. Thus, HDPE matrix reinforced with hybrid MWCNT/Al_2_O_3_ was found to be a favorable bio-composite for made-up joints. The dispersion of hybrid MWCNT/Al_2_O_3_ contributes to reducing the cytotoxic activity and water absorption, while the tensile strength, breaking percentage, and hardness improved impressively [41].

Previous studies have shown that HDPE is biocompatible, non-toxic, and has excellent biological properties, and that carbonaceous nanofillers of different sizes, like graphene oxide (Gr) sheets, encourage osteoblast adhesion, growth, and differentiation. In many tissue engineering uses, carbon-based nanomaterials are added to polymers to enhance biocompatibility. Additionally, osseointegration and bioactivity can be enhanced by titanium dioxide (TiO_2_). The study’s objective was to assess the integration of composite TiO_2_/graphene nano-fillers into the HDPE matrix for the first time. TiO_2_ NPs and graphene hybrid nanofillers were investigated in loading quantities of 0.5%, 1.0%, 1.5%, and 2.0 wt.%. To determine the effects of incorporating hybrid nanofillers into the HDPE matrix, the structure of nanocomposites, water absorption, mechanical performance, and tribological performance were investigated.

## 2. Materials and Methods

The current study targeted High-Density Polyethylene (HDPE) as a base material. HDPE powder supplied by Sigma Aldrich Co. (Paris, France) had a specification of 0.94 gm/cm^3^ as density, 27 MPa tensile stress, and a particle size average of 40:90 μm. The matrix reinforced by hybrid nanofillers consists of titanium oxide (TiO_2_) NPs and nano-graphene (Gr), which were purchased from US Research Nanoparticles (Houston, TX, USA). TiO_2_ NPs had 95.5% purity, a size of 40 nm for spherical particles, and a specific surface area of 35 m^2^/g. Nano-graphene with 95% purity had 3–6 layers with a thickness of 2–8 nm and a specific surface area of 500–1000 m^2^/g.

Assuring uniform dispersion of the fillers through the HDPE matrix is a significant challenge during sample preparation. To correlate the dispersion and prevent agglomeration, ethanol was confirmed as the solvent. Hybrid nanofillers were added to ethanol solvent and stirred for 10 min at 200 rpm. The mixture is then stirred for 15 min with Dihan, HG-15, Vietnam, to improve particle dispersion. The mixture is added to the HDPE matrix, and the composite is re-stirred in two stages: first with a rotating stirrer at 600 rpm for 10 min, then with a Dihan, HG-15, Beijing, China, for 15 min. For 40 min, the resin is extruded through the cylindrical copper mold and pressed with 25 MPa at temperatures up to 200 °C. The heating temperature helped the solvent (ethanol) to evaporate. The presence of hybrid nanofillers of TiO_2_ NPs and Gr (in a 1:1 ratio) in samples with loading contents of 0.5%, 1.0%, 1.5%, and 2.0 wt.% was confirmed. Table 1 shows the labs that tested the samples. 

## 3. Experiments Details

Several procedures were adopted to analyze the physical, mechanical, and tribological features of HDPE nanocomposites. On this study IR spectroscopy was adopted to identify the components’ frequencies in the interaction bond of the different loading content of HDPE matrix. The experiments were carried out via a Beckman IR 4250 spectrophotometer, USA. Moreover, IR results were confirmed with a wavenumber range of 400–2000 cm^−1^ and transmittance (%).

The mechanical properties were evaluated to obtain the tensile stress, elastic modulus, breaking stress percentage, and hardness. The HDPE nanocomposites were examined using uniaxial universal testing DFM-300KN, China, according to ASTM standard D1621. The hardness test was evaluated using the Durometer device, Shore D, according to ASTM standard D2240. The hardness test was conducted for five different positions, and some discrepancies were observed, so the average hardness was determined. Structural properties and phase identification of HDPE nanocomposites were examined via the Siemens D500 X-ray Diffractometer, Germany. X-ray diffraction (XRD) analysis was carried out within a measurement range of 0° to 70° (2θ), while the test was configured at a current of 30 mA and voltage of 30 kV.

Based on ASTM D750-95, the relative water absorption was determined when the samples were completely immersed in distilled water for 72 h at room temperature. The water absorption test was performed by weighing the samples before and after immersing them in water.

The friction and wear mechanisms were evaluated using a pin-on-disc tribometer based on ASTM standard G99-95. The samples were examined against a stainless-steel alloy disk under dry sliding conditions at a relative humidity of 60% and 30 °C. The friction coefficient, COF, of the samples is evaluated under a specific weight load of 2, 4, 6, 8, and 10 N and a sliding velocity of 0.1 m/s. While the weight loss was estimated at various sliding distances of 31.4, 47.1, 62.8, 94.2, and 125.6 m under an applied load of 10 N. Samples are usually weighed before and after testing to calculate weight loss ∆m, gm. With the knowledge of the sliding distance L, material density ρ, and applied load F_n_, the wear rate, WR, was calculated using the formula WR = ∆m⁄(L_ρ_ × F_n_). The topography of the worn surfaces was analyzed using an electronic microscope (OLYMPUS BX53M, Tokyo, Japan) and SEM microscope (JCM-6000Plus; JEOL, Tokyo, Japan).

## 4. Results

The HDPE samples reinforced by hybrid nanofillers with a ratio of 1:1 of TiO_2_ NPs and Gr were characterized by IR-spectroscopy and X-ray diffraction. The IR spectra analysis of HDPE nanocomposite samples is illustrated in Figure 1a. The collected data from acquired spectra contributes to evaluating the interaction between HDPE resin and the hybrid nanofiller. In the case of pure HDPE, Sample O, the transmittance exhibited bands at wave numbers of 815, 1118, 1461, 1793, 2861, and 2973 cm^−1^ [42,43,44]. The status and peak intensities of the bands may be assigned to various groups of bending vibrations. The functional bonds can be identified as single, double, or non-conjugated. The analysis indicates that the double cis bond (-CH=CH-) is seen at 815 cm^−1^, while the double trans bond (-CH=CH_2_) appears at 1118 cm^−1^. Moreover, the CH_3_ vibrational bond is observed at wave number 1461 cm^−1^ and the CH_2_ functional bond is located at 2861 cm^−1^. IR peak at wave number 1793 cm^−1^ indicates a functional non-conjugated group. While a functional single bond group (C-CH_3_) appears at 2973 cm^−1^. IR spectra were recorded for HDPE nanocomposite samples, and the results indicate that the samples provide the same functional band groups as the sample O. These results make it evident that the hybrid nanofillers were successfully incorporated into an HDPE matrix.

The X-ray diffraction pattern of HDPE hybrid nanocomposites is provided in Figure 1b. The XRD pattern of pure HDPE had two broad peaks at around 2θ of 21.4°, with high intensity, and 24.2°, with low intensity. These two peaks provide Bragg reflection planes (110) and (200), respectively [45]. The XRD patterns of the samples loaded with hybrid nanofillers that exhibited compatibility and a semi-crystalline pattern with the pure sample were acquired. It was observed that hybrid nanofiller materials, TiO_2_ NPs and Gr, have no negative effect on the structural crystal features of HDPE. This confirms that no chemical reaction occurred between the matrix and the fillers. Accordingly, it has been observed that the XRD pattern results of HDPE hybrid nanocomposites are consistent with those in the previous literature [46,47].

Water absorption is a basic indicator for evaluating the performance of nano-polymers because it affects their life cycle. Figure 2 demonstrates the water absorption percentage of pure HDPE and its nanocomposite samples. It was indicated that HDPE nanocomposite samples exhibited lower water absorption by about 39–77% compared to that of pure HDPE. It may be realized that the loading of hybrid nanofiller had a significant effect on the composite’s structural features. A good physical mixture between hybrid nanofillers and HDPE matrix contributes to improving wettability and reducing voids. This could be evident from the fact that increasing the loading of hybrid nanofillers leads to reduced water accumulation through the internal structure of the nanocomposite matrix [41].

The mechanical properties of HDPE nanocomposite samples were evaluated by examining the tensile strength, elastic modulus, and hardness. The stress-strain curve for HDPE nanocomposite samples was illustrated in Figure 3a. It can be observed that the addition of hybrid nanofillers leads to an increase in ultimate tensile stress. The maximum tensile stress is achieved with a higher loading amount of hybrid nanofiller, Sample D, resulting in a tensile stress of 28.4 MPa, as displayed in Figure 3b. Moreover, the strain of the nanocomposite samples was enhanced by increasing the filler loading content. It is evident from the fact that the fillers limit the interspaces between molecules of the HDPE matrix, which leads to restricted mobility and plastic deformation [48]. Sample B shows the highest strain compared with other loading amounts. The hardness values of HDPE nanocomposite samples are presented in Figure 3b. It can be found that the hardness increases linearly with increasing the loading content of hybrid nanofillers. The hardness Shore D value increases up to 20% with the highest loading content of filler compared to pure HDPE. The improvement percentages of tensile strength and elastic modulus were estimated, as illustrated in Figure 3c,d, respectively. High-loading content exhibits the highest improvement percentages of tensile strength and elastic modulus, with approximately 19.8% and 9.4% improvement, respectively. This shows that nanofillers contribute to enhancing the ductility and strain rate of HDPE matrices [36,40]. Table 2 outlines the mechanical features of HDPE nanocomposite samples compared with the pure sample.

The friction coefficient and wear rate were measured in accordance with ASTM G99-95 for HDPE nanocomposite samples. Figure 4a shows the variation of the friction coefficient against the normal applied loads for different loading contents of hybrid nanofillers, TiO_2_ NPs, and Gr. For all HDPE samples, the data indicate a significant rise in the friction coefficient because of the increased effective loads. As for samples embedded with hybrid nanofillers, it can be assumed that the nanofillers contribute to reducing friction in the contact area. This may be attributed to the rolling effect of TiO_2_ NPs and the self-lubricating feature of graphene, which exhibit a distinct impact on the sliding conditions. Hence, it can be concluded that increasing loading content of hybrid nanofillers leads to a gradual decrease in the friction coefficient [49,50]. This note is considered consistent even with a loading content of 1.5 wt.% of hybrid nanofillers and an equal ratio of TiO_2_ NPs and Gr. The minimum friction coefficient is achieved with Sample C and a 1.5 wt.% loading amount of hybrid nanofillers, so the friction coefficient reduces with approximately a 27.5% improvement. An increase in friction coefficient may be attributed to high loading amount of hybrid nanofillers in the polymer matrix, as occurred with Sample D, which increases the possibility of particle agglomeration. It is evident that the agglomeration response is to restrict the particles’ dispersion and the shear flows between layers. Furthermore, the wear rate was estimated via the experimentally recorded amount of sample weight that indicated the weight loss of each sample, as shown in Figure 4b. The results indicate that the same trend in friction behavior is repeated even with the wear rate. It seems clear from this figure that the reinforcement of the HDPE matrix with hybrid nanofillers exhibits a good reaction to wear. Under the sliding action, the wear rate for Sample C, with a loading content of 1.5 wt.%, reduces with an overall decrease of 36.3% less than the pure HDPE. This improvement can be attributed to the presence of graphene, which contributes to the formation of a lubricant layer on the contact area [39,50]. In addition, the rolling effect of particles reduces friction, which enhances wear performance. While many problems appear at the interface with the overloading of nanofillers like particle agglomeration and poorly bonded slipping layers. This may be attributed to the high loading content of over 1.5 wt.% of nanofillers, which is thought to constrain the internal molecule’s mobility, thereby causing plastic deformation. This shows that loading content of 1.5 wt.% of hybrid nanofiller contributes to enhancing the frictional and wear rates of HDPE matrix.

The study of worn surfaces aims to elucidate the damage and deformation mechanisms of sliding surfaces via optical images, topography, and SEM micrographs. The topography of worn surfaces was evaluated using optical microscopic techniques with 2D and 3D scanned images, as demonstrated in Figure 5. It is clearly visible in the pure sample O that the worn surface appears damaged as the plastic deformation layers spread wear tracks and plow on the contact area. The topography images of nanocomposites samples indicate the good dispersion of hybrid nanofillers inside an HDPE matrix that leads to reduced particle agglomeration and effectively separates the sliding interfaces. It can be observed that the surface of Sample A seems worn, with the appearance of grooves and wear tracks and a narrow plowing area, which evidences the weight loss being less compared with pure HDPE. Additionally, the increase in loading content is still a positive indicator, as the surface of sample B appears less damaged than that of sample A. It can be demonstrated that the surface of Sample C appears smoother with fewer cracks, grooves, and wear marks. On the contrary, wear marks and plowing areas reappear on the surface of the sample D. It can be assumed that an HDPE matrix containing hybrid nanofillers, up to a loading content of 1.5 wt.%, leads to the formation of a self-lubricating film and thus a low wear rate. While continuing to add more than this limit is considered inappropriate [51]. This may affect friction and wear performance due to agglomeration and incoherence.

The SEM images were employed to inspect and examine the mechanisms of the matrix and the nanocomposite structure, as depicted in Figure 6. It can be observed that the surface of Sample O seems to weaken the cohesion layers, where cracks and furrows appear. Moreover, the samples reinforced with hybrid nanofiller exhibit good cohesion between interlayers and less breakup. The image recorded for sample A indicates the presence of furrows and voids on the surface, unless the sample structure looks better compared to the pure sample. It can be observed that the matrix structure improves with the increase in content loading, as voids and cracks are limited, as illustrated in SEM images of samples B and C. This reveals that the hybrid nanofillers, TiO_2_ NPs and Gr, contribute to enhancing the matrix structure and bonding the layers [50]. Therefore, the incorporation of nanofillers with the matrix prevents crack propagation on the sample surface. Nevertheless, increasing the nanofiller amount causes agglomeration and weakens the dispersion of particles, which leads to plastic flow at the sliding area.

## 5. Conclusions

HDPE nanocomposites reinforced with hybrid nanofillers, TiO_2_ NPs, and graphene were investigated for hip joint prosthetic replacement. Samples included mixed nanofillers with fracture weights of 0.5, 1.0, 1.5, and 2.0%. All samples were tested to determine the nanofiller dosage for optimal performance. IR spectra and XRD patterns were used to characterize the HDPE matrix. The nanofillers were successfully integrated into the HDPE matrix without affecting its crystal structure. Hybrid nanofillers, TiO_2_ NPs and Gr, were compatible with HDPE matrix. The good dispersion of hybrid nanofillers reduced water uptake by 77% compared to pure HDPE.

Hybrid nanofiller filling increased mechanical characterizations almost linearly. The HDPE nanocomposite sample enhanced with 2 wt.% TiO_2_ NPs and Gr, 1:1, had increased tensile strength, elastic modulus, and hardness by 19.8%, 9.4%, and 20%, respectively. Estimating friction coefficient and wear rate assessed HDPE nanocomposites’ tribological performance. Samples with 1.5 wt.% hybrid nanofillers had the lowest friction coefficient and wear rate, reducing them by 27.5% and 36.3%, respectively. Hybrid nanofillers are distinguished by their single duty. TiO_2_ NPs reduce frictional shear force by rolling, while graphene forms a lubricating layer to separate and shield sliding surfaces. Particle agglomeration at higher loadings (2.0 wt.% hybrid nanofillers) restricts layer motion, raising plastic deformation and the fractional effect. Optical topography and SEM photos showed the HDPE nanocomposites matrix structure from worn surfaces. HDPE reinforced with 1.5 wt% hybrid nanofillers had a good matrix structure with smoother surfaces and low gaps. The matrix’s uniform nanofiller spread prevents crack propagation. However, for sample with 2.0 wt.% hybrid nanofillers loading, it can confirm that the plastic flow and plowing area at the contact zone may be due to particles agglomeration and weakening dispersion.

Finally, this study reveals that HDPE nanocomposites with 1.5 wt.% hybrid nanofillers, TiO_2_ NPs, and graphene in equal ratios could be recommend using as a hip joint replacement material. Furthermore, enhancing mechanical and tribological performance of HDPE matrix yields a material with a long lifespan and low wear rate.

## Figures and Tables

**Figure 1 jfb-14-00140-f001:**
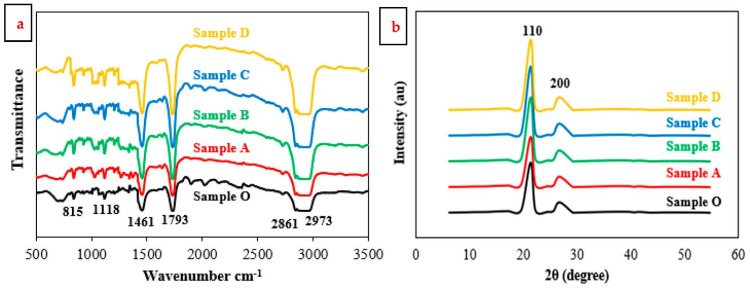
(**a**) IR spectra of HDPE nanocomposite samples; (**b**) XRD patterns of HDPE nanocomposite samples.

**Figure 2 jfb-14-00140-f002:**
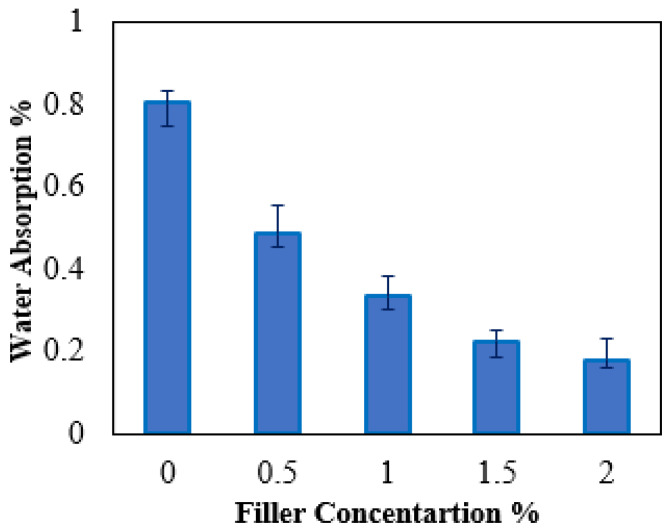
Water absorption percentage of HDPE nanocomposite samples.

**Figure 3 jfb-14-00140-f003:**
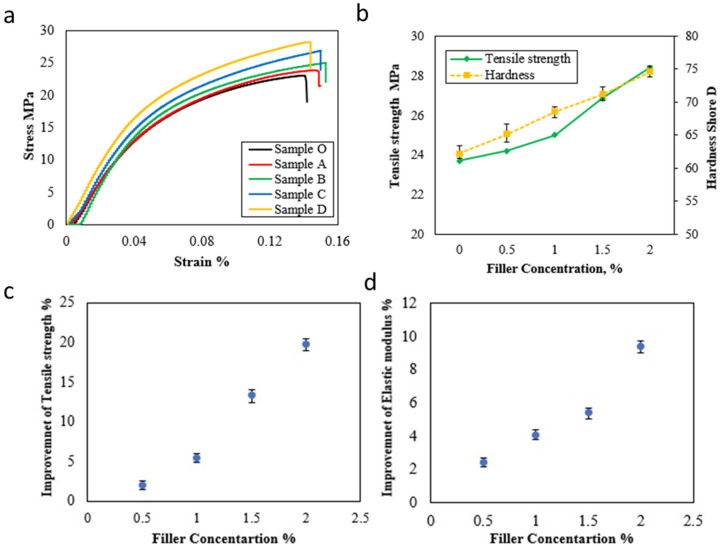
(**a**) Stress-strain curve of HDPE nanocomposite samples, (**b**) Tensile strength and hardness Shore D of HDPE nanocomposite samples, (**c**) Improvement of tensile strength of HDPE nanocomposite samples (**d**) Improvement of elastic modulus of HDPE nanocomposite samples.

**Figure 4 jfb-14-00140-f004:**
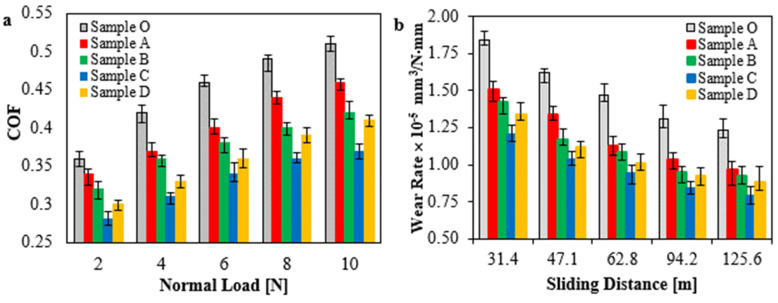
(**a**) Friction coefficient of HDPE nanocomposite sample, and (**b**) Wear rate of HDPE nanocomposite samples.

**Figure 5 jfb-14-00140-f005:**
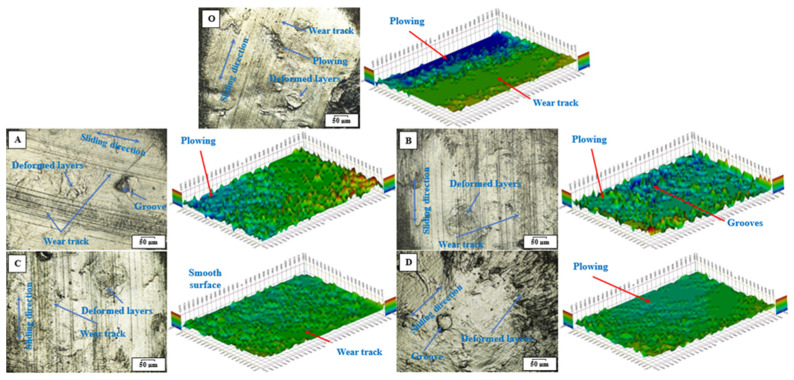
Optical images of 2D and 3D topography of worn surfaces of HDPE nanocomposite samples, i.e., sample O, sample A, sample B, sample C, and sample D.

**Figure 6 jfb-14-00140-f006:**
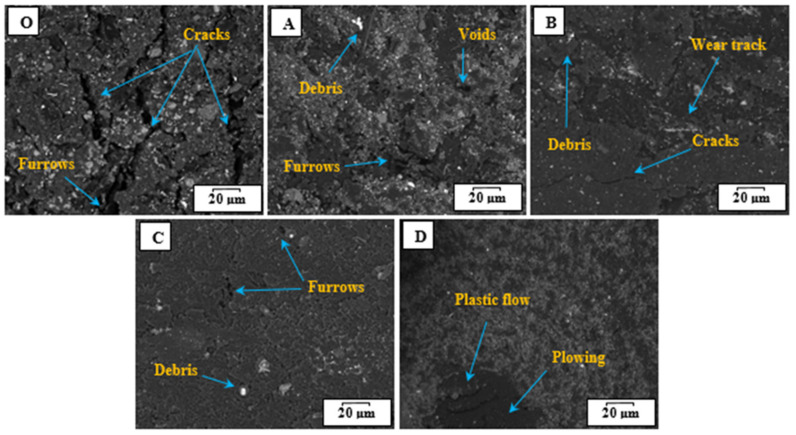
SEM images of worn surfaces of HDPE nanocomposite samples., sample O, sample A, sample B, sample C, and sample D.

**Table 1 jfb-14-00140-t001:** Various sample compositions of HDPE/Hybrid nanocomposites.

Sample No	HDPE	TiO_2_ NPs	Gr
Sample O	100%	-	-
Sample A	99.5%	0.25%	0.25%
Sample B	99%	0.5%	0.5%
Sample C	98.5%	0.75%	0.75%
Sample D	98%	1.0%	1.0%

**Table 2 jfb-14-00140-t002:** Mechanical properties of HDPE nanocomposite samples.

Sample No	Tensile Strength (MPa)	Elastic Modulus (GPa)	%Breaking Strain
Sample O	23.4 ± 2.7	409.4 ± 6.1	13.6 ± 1.3
Sample A	24.2 ± 2.5	419.4 ± 6.9	14.4 ± 1.6
Sample B	25.0 ± 2.6	425.9 ± 7.3	14.9 ± 1.5
Sample C	26.8 ± 3.3	431.5 ± 6.3	14.4 ± 1.9
Sample D	28.4 ± 3.1	447.9 ± 8.1	14.1 ± 2.1

## Data Availability

All data used in this study are declared in the paper.

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
