# Peer review of "Mechanical and Tribological Performance of HDPE Matrix Reinforced by Hybrid Gr/TiO2 NPs for Hip Joint Replacement"

_jfb, 2023, doi:10.3390/jfb14030140_

Round 1

Reviewer 1 Report

1. Scientific aspects. From the general scientific perspective, the concrete usefulness of the presented materials in relation to the biological parameters must be more clearly highlighted.

2. Regarding the quality of the presentation, the text must (re)written more carefully, figures must have a unitary format and separated by an explanatory text and not placed successively (like Fig 4 and Fig 5, Fig 8 and Fig 9) and must be commented in the text (should not end a section without a text) etc.

Author Response

Reply to reviewers:

Manuscript ID: jfb-2206189

Manuscript Title: Mechanical and Tribological Performance of HDPE matrix Reinforced by Hybrid Gr/TiO2 NPs for hip joint replacement

 The authors would like to thank the editor and reviewers for giving us the opportunity to submit a revised draft of the manuscript “Mechanical and Tribological Performance of HDPE matrix Reinforced by Hybrid Gr/TiO2 NPs for hip joint replacement” for publication in the Journal of Functional Biomaterials. We appreciate the time and effort that you and the reviewers dedicated to review and analyze our manuscript and are grateful for the valuable comments on and useful remarks to our paper. We have considered most of the suggestions made by the reviewers. Those modifications are highlighted in the manuscript. Please see below, for a point-by-point response to the reviewers’ comments and concerns.

Reviewer 1

Comments and Suggestions for Authors

  1. Scientific aspects. From the general scientific perspective, the concrete usefulness of the presented materials in relation to the biological parameters must be more clearly highlighted.

 Response: Thank you for pointing this out. The reviewer is correct, and we have clarified that relation in the introduction.

  1. Regarding the quality of the presentation, the text must (re)written more carefully, figures must have a unitary format and separated by an explanatory text and not placed successively (like Fig 4 and Fig 5, Fig 8 and Fig 9) and must be commented in the text (should not end a section without a text) etc.

Response: we revised and combined all figures in the manuscript and commented in the text.

Reviewer 2 Report

This work is interesting and well done. There are a few language errors. It can be accepted after the langaage is checked.

Author Response

Reply to reviewers:

Manuscript ID: jfb-2206189

Manuscript Title: Mechanical and Tribological Performance of HDPE matrix Reinforced by Hybrid Gr/TiO2 NPs for hip joint replacement

Reviewer 2

Comments and Suggestions for Authors

This work is interesting and well done. There are a few language errors. It can be accepted after the language is checked.

Response: Thank you, Modified.

Text revised and modified at manuscript, and English of the manuscript was revised and to prevent any grammar errors, the manuscript was also revised using Grammarly software.

Reviewer 3 Report

This paper presents the Mechanical and Tribological Performance of HDPE matrix Reinforced by Hybrid Gr/TiO2 NPs for hip joint replacement.

The following points need to be clarified:

It is suggested that the authors should modify last part of Introduction to: 1) clearly mention the goal and novelty of this work, 2) mention the methodology used and its important to place the major hypothesis.

The skeleton of the paper could be summarized at the end of Section 1.

The paper lacks a discussion on the weakness and limitations of the present methodology/study.

The conclusion of the present work is too long. Only main findings should be briefly placed.

Fig. 5 and Table 2 do not show the error bars. That needs to be addressed.

Author Response

Reply to reviewers:

Manuscript ID: jfb-2206189

Manuscript Title: Mechanical and Tribological Performance of HDPE matrix Reinforced by Hybrid Gr/TiO2 NPs for hip joint replacement

Reviewer 3

The following points need to be clarified:

It is suggested that the authors should modify last part of Introduction to:

  • clearly mention the goal and novelty of this work,

Response: we have modified in the last part of introduction in paper

  • mention the methodology used and its important to place the major hypothesis.

ResponseAfter stirring the nanocomposite sample, the major hypothesis and final shape were extruded.

Assuring uniform dispersion of the fillers through the HDPE matrix is a significant challenge during sample preparation. To correlate the dispersion and prevent agglomeration, ethanol was confirmed as the solvent. For 10 minutes, hybrid nanofillers were added to an ethanol solvent and stirred at 200 rpm. For 15 minutes, the mixture is stirred to improve particle dispersion. The mixture is added to the HDPE matrix, and the composite is re-stirred in two stages, first with a rotating stirrer at 600 rpm for 10 minutes, then with a Dihan, HG-15, China, for 15 minutes. For 40 minutes, the resin is extruded through the cylindrical copper mould and pressed with 25 MPa at temperatures up to 200°C. The heating temperature aids in the evaporation of the solvent ethanol. The presence of hybrid nanofillers of TiO2 NPs and Gr (in a 1:1 ratio) in samples with loading contents of 0.5%, 1.0%, 1.5%, and 2.0 wt.% was confirmed.

  • The skeleton of the paper could be summarized at the end of Section 1.

Response: Done

Previous studies have shown that HDPE is biocompatible, non-toxic, and has excellent biological properties and that carbonaceous nanofillers of different sizes, like graphene oxide (Gr)sheets, encourage osteoblast adhesion, growth, and differentiation [. In many tissue engineering uses, carbon-based nanomaterials are added to polymers to enhance biocompatibility. Additionally, osseointegration and bioactivity can be enhanced by titanium dioxide (TiO2). The study's objective was to assess the integration of composite TiO2/graphene nano-fillers into the HDPE matrix for the first time. TiO2 NPs and graphene hybrid nanofillers were investigated in loading quantities of 0.5%, 1.0%, 1.5%, and 2.0 wt.%. To determine the effects of incorporating hybrid nanofillers into the HDPE matrix, the structure of nanocomposites, water absorption, mechanical, and tribological performance were investigated

  • The paper lacks a discussion on the weakness and limitations of the present methodology/study.

Response: Because our study is the first to discuss the composition of hybrid TiO2/graphene nano-fillers in the HDPE matrix in order to assess mechanical and tribological properties, it paves the way for many others to investigate and improve. However, the majority of the principles were gathered by our research method.

  • The conclusion of the present work is too long. Only main findings should be briefly placed.

Response: Done

HDPE nanocomposites reinforced with hybrid nanofillers, TiO2 NPs and graphene, are being investigated for hip joint prosthetic replacement. Samples included mixed nanofillers with fracture weights of 0.5, 1.0, 1.5, and 2.0%. All samples were tested to determine nanofiller dosage for optimal performance. IR spectra and XRD patterns characterised HDPE matrix. The nanofillers were successfully integrated into HDPE matrix without affecting its crystal structure. Hybrid nanofillers, TiO2 NPs and Gr, were compatible with HDPE matrix. The good dispersion of hybrid nano-fillers reduces water uptake by 77% compared to neat HDPE.

Hybrid nanofiller filling increased mechanical characterizations almost linearly. The HDPE nanocomposite sample enhanced with 2 wt.% TiO2 NPs and Gr, 1:1, increased tensile strength, elastic modulus, and hardness by 19.8%, 9.4%, and 20%, respectively. Estimating friction coefficient and wear rate assessed HDPE nanocomposites' tribological performance. Samples with 1.5 wt.% hybrid nanofillers had the lowest friction coefficient and wear rate, reducing them by 27.5% and 36.3%, respectively. Hybrid nanofillers are distinguished by their single duty. TiO2 NPs reduce frictional shear force by rolling while graphene forms a lubricating layer to separate and shield sliding surfaces. Particle agglomeration at higher loading, 2.0 wt.% hybrid nanofillers, restricts layer motion, raising plastic deformation and fractional effect. Optical topography and SEM photos showed the HDPE nanocomposites matrix structure from worn surfaces. HDPE reinforced with 1.5 wt% hybrid nanofillers has a good matrix structure with smoother surfaces and low gaps. The matrix's uniform nanofiller spread prevents crack propagation. However, finding plastic flow and ploughing at the sliding area due to particle agglomeration and weaken dispersion at 2.0 wt.% hybrid nanofillers loading.

HDPE nanocomposites with 1.5 wt.% hybrid nanofillers, TiO2 NPs, and graphene in equal ratios may be used for hip joint replacement. Enhancing mechanical and tribological performance yields a material with a long lifespan and low wear rate.

  • 5 and Table 2 do not show the error bars. That needs to be addressed.

Response: Done

Reviewer 4 Report

Dear authors,

The manuscript entitled "Mechanical and Tribological Performance of HDPE matrix Reinforced by Hybrid Gr/TiO2 NPs for hip joint replacement" is an original article, which is interesting for the readers of the Journal of Functional Biomaterials. 

Composite materials based on HDPE, TiO2 nanoparticles and nano-graphene with various filler content were evaluated in terms of composition and structure by FTIR and XRD which showed successful incorporation of the nanoparticles in the HDPE matrix. Water absorption on the obtained composites showed a decreasing trend versus increasing filler content. Mechanical characterizations showed an increase in the properties versus the loading content of hybrid nanofillers. The article is well-written and organized, and the figures are well-explained and illustrated. The references are relevant to the study and the overall merit of the manuscript is high. 

I recommend accepting the article after minor revision. The chemical formulas should be written with subscript.

Author Response

Reply to reviewers:

Manuscript ID: jfb-2206189

Manuscript Title: Mechanical and Tribological Performance of HDPE matrix Reinforced by Hybrid Gr/TiO2 NPs for hip joint replacement

Reviewer 4

The manuscript entitled "Mechanical and Tribological Performance of HDPE matrix Reinforced by Hybrid Gr/TiO2 NPs for hip joint replacement" is an original article, which is interesting for the readers of the Journal of Functional Biomaterials. 

Composite materials based on HDPE, TiO2 nanoparticles and nano-graphene with various filler content were evaluated in terms of composition and structure by FTIR and XRD which showed successful incorporation of the nanoparticles in the HDPE matrix. Water absorption on the obtained composites showed a decreasing trend versus increasing filler content. Mechanical characterizations showed an increase in the properties versus the loading content of hybrid nanofillers. The article is well-written and organized, and the figures are well-explained and illustrated. The references are relevant to the study and the overall merit of the manuscript is high. 

I recommend accepting the article after minor revision. The chemical formulas should be written with subscript.

Response: Thank You, Done.

Round 2

Reviewer 3 Report

The critics were addressed properly.